# Intramolecular dynamic coupling slows surface relaxation of polymer glasses

Houkuan Tian [1,7], Jintian Luo [1,7], Qiyun Tang [2] ✉, Hao Zha[1], Rodney D. Priestley [3] ✉, Wenbing Hu [4] & Biao Zuo [1,5,6] ✉

Over the past three decades, studies have indicated a mobile surface layer with steep gradients on glass surfaces. Among various glasses, polymers are unique because intramolecular interactions – combined with chain connectivity – can alter surface dynamics, but their fundamental role has remained elusive. By devising polymer surfaces occupied by chain loops of various penetration depths, combined with surface dissipation experiments and Monte Carlo simulations, we demonstrate that the intramolecular dynamic coupling along surface chains causes the sluggish bulk polymers to suppress the fast surface dynamics. Such effect leads to that accelerated segmental relaxation on polymer glass surfaces markedly slows when the surface polymers extend chain loops deeper into the film interior. The surface mobility suppression due to the intramolecular coupling reduces the magnitude of the reduction in glass transition temperature commonly observed in thin films, enabling new opportunities for tailoring polymer properties at interfaces and under confinement and producing glasses with enhanced thermal stability.

In the same way, the surfaces of crystals are not always crystal-like, the surfaces of glasses are not so glassy[1–15]. Atoms and molecules near the free surface of a glass are in the mobile state. The accelerated surface dynamics are the foundation of many physical processes, such as surface dissipation[16,17], interfacial contacts, and adhesion[18,19], and are relevant to distinct physical properties, including physical aging[20,21], stability[22–24], crystallization[25,26], yield, and failure[27–29] of devices at the nanoscale. Contrary to the general understanding of glass physics, in which the dynamics of molecules are spatially correlated as a result of cooperative relaxation[30,31], the enhanced surface dynamics in some glasses with simple molecular geometry are not coupled with relaxations in the underlying bulk[8,12,13]. In particular, Zhang and Fakhraai disclosed that the surface diffusion coefficients of ordinary and ultra-stable molecular glasses of $N,N'$-bis(3-methylphenyl)-$N,N$

'-diphenylbenzidine are almost undistinguished although the bulk relaxation times differ by about 13–20 orders of magnitude[8]. The dynamical decoupling can be attributed to the removal of intermolecular cooperativity at surfaces[32,33]. Furthermore, machine learning and simulations have shown that the fast surface dynamics are disconnected from the microscopic structure and local density near surface[13,34,35]. Such weak surface–bulk and dynamic–structure correlations hinder the ability to tailor the surface relaxation of materials by manipulation of the glass structures, i.e., the general ways to alter the glass dynamics.

Polymers behave differently than simple molecules because of the added complexity of chain connectivity, which may alter their dynamics owing to the intramolecular cooperativity arising from intramolecular barriers for dihedral motion along chain molecules[36–39].

[1]School of Chemistry and Chemical Engineering, Key Laboratory of Surface & Interface Science of Polymer Materials of Zhejiang Province, Zhejiang Sci-Tech University, Hangzhou 310018, China. [2]Key Laboratory of Quantum Materials and Devices of Ministry of Education, School of Physics, Southeast University, Nanjing 211189, China. [3]Department of Chemical and Biological Engineering, Princeton Institute for the Science and Technology of Materials, Princeton University, Princeton, NJ 08544, USA. [4]Department of Polymer Science, School of Chemistry and Chemical Engineering, State Key Lab of Coordination Chemistry, Nanjing University, Nanjing 210023, China. [5]Zhejiang Provincial Innovation Center of Advanced Textile Technology, Shaoxing 312000, China. [6]Zhejiang Sci-Tech University Shengzhou Innovation Research Institute, Shengzhou 312400, China. [7]These authors contributed equally: Houkuan Tian, Jintian Luo. ✉e-mail: qtang@seu.edu.cn; rpriestl@princeton.edu; chemizuo@zstu.edu.cn

Segmental vibrations, bond rotations, and other local intramolecular motions can propagate dynamics along polymer backbones[40], thereby building up a large-scale $\alpha$-relaxation in polymer glasses. The intramolecular dynamic coupling has recently been employed to provide a unified explanation for the chain length dependence of glass transition temperature ($T_g$) in bulk polymers[36]. An alternative approach exclusively describing the intramolecular coupling is the sliding model proposed by de Gennes[41,42], which assumes that the randomly generated kinks or free volume on a chain molecule can diffuse along its backbone and activate the cooperative segmental dynamics. Such an approach has been applied widely in theories[43,44] and simulations[45,46] to study the glass transition[43,44], structural relaxation[45], and crystallization kinetics[46] of confined polymers, demonstrating a strong association between intramolecular coupling and the various dynamic processes. In particular, for polymers near surfaces, in which a chain molecule spans the whole or partial surface-mobility gradients[9], the intramolecular dynamical correlations could play a part in facilitating the propagation of enhanced surface dynamics and ensuring the smooth transition near surfaces. However, how intramolecular coupling influences the surface gradient dynamics of polymer glasses is not clear.

Herein, we explored the effects of intramolecular dynamic coupling on the surface relaxation of polymers by designing polymer surfaces with chain loops of different sizes and various penetration depths. Our results demonstrate that accelerated surface relaxation was suppressed by extending the surface chain loops deeper into the slow-relaxing film interior. This observation confirms that the surface and bulk dynamics of polymer glasses are indeed affected by each other because of the dominant dynamic coupling along surface chains, unlike the decorrelation of surface and bulk dynamics occurring in the simple glass formers[8,12,13,32]. The surface-loop-assisted surface-bulk dynamical correlations suggest a way to tailor the $T_g$ and structural relaxation of polymers at interfaces and under confinement by changing the conformation of surface chains.

## Results

### Generation of surface loops via experiments and simulations

We made use of the selective surface segregation of surface energy–distinct components in a statistical random copolymer of poly(methyl methacrylate-*sta*-pentafluorostyrene) [P(MMA-*sta*-PFS); the surface free energies of the MMA and PFS homopolymers are 45.7 and 20.3 mN/m, respectively; see Supplementary Table 1 in the Supplementary Information (SI)] to generate the loop structure both experimentally and in simulations. In experiment, a series of P(MMA-*sta*-PFS) with various mole fractions of PFS ($f_{PFS}$) was prepared via atom transfer radical polymerization (see the synthesis details in the SI). The copolymers were introduced as additives into a PMMA matrix using a solution blend method, and the polymer films were prepared by spin-casting the solutions. The films were annealed at $T_g + 35$ K, which allowed the spontaneous segregation of PFS with lower surface free energy and thus greater affinity to the air interface, as well as concomitant surface depletion of MMA with higher surface energy. Surface analyses (i.e., measurements of the water contact angle, X-ray photoelectron spectroscopy, and interfacial sum frequency generational spectroscopy[47]) of the film surface confirmed this process and might indicate the formation of various chain loops at the surface (Supplementary Fig. 4).

We performed dynamic Monte Carlo (MC) simulations[45,46] based on the coarse-grained models of the same experimental systems (see the modeling details in Section 6 in the SI) to assess the local conformation of surface chains. Thermal equilibration of the simulated film at a high temperature resulted in the enrichment of PFS towards the free surface (Supplementary Fig. 5). Figure 1A and Inset of Fig. 1B show the distribution of PFS near the surface region and a snapshot of a simulated surface P(MMA-*sta*-PFS) chains. More than 85% surface chains with PFS located on the surface form loops (Supplementary Table 3). Inset of Fig. 1B depicts typical Gaussian-like surface loops formed by polymer strands between the surface-anchoring PFS units. Statistically, the surface loops penetrated deeper into the films mixed with copolymers with larger $f_{PFS}$ (Fig. 1B).

The fraction of PFS units at the surface and the number of surface loops increased as more P(MMA-*sta*-PFS) was added to the PMMA matrix; Supplementary Fig. 6. The effective surface areal fraction of PFS ($C_{PFS}$) derived from the contact angle values and by the MC simulation increased with P(MMA-*sta*-PFS) weight fraction ($X_c$) in the films before reaching saturation; Fig. 1C. To investigate the effects of the conformation of the surface chain exclusively, the surface

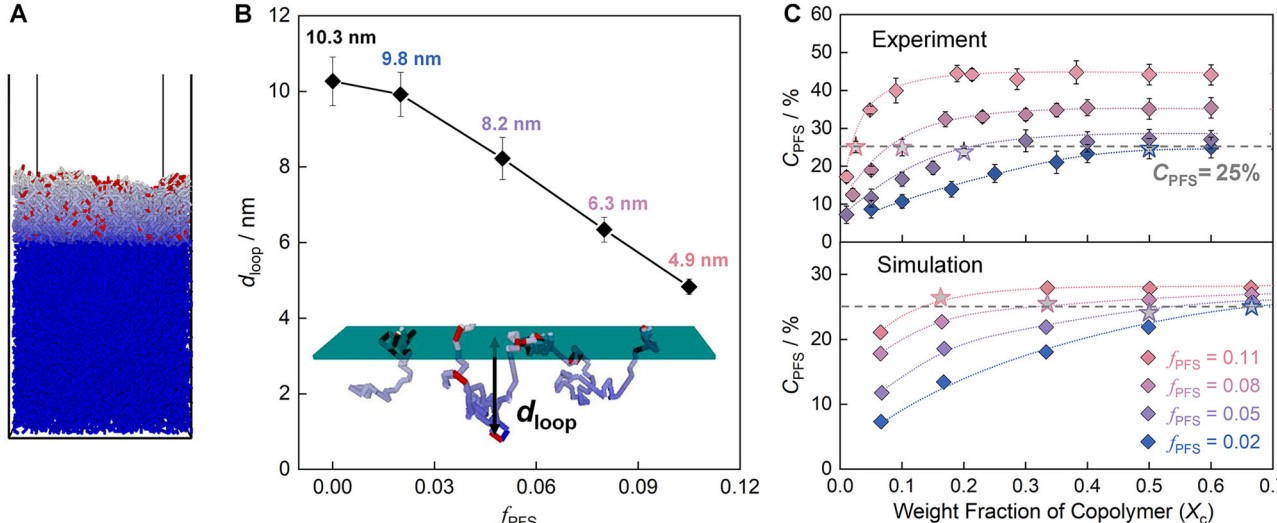

**Fig. 1 | Loop structure at the surface of PMMA/P(MMA-*sta*-PFS) films. A** The simulated snapshot of the well-annealed PMMA/P(MMA-*sta*-PFS) films; red and blue rods represent PFS and MMA units, respectively; the bulk area of the films is displayed in blue. **B** Average surface loop depths ($d_{loop}$) extending in films with a $C_{PFS}$ of 25% and various $f_{PFS}$. The inset in (**B**) shows a snapshot of a P(MMA-*sta*-PFS) chain at film surfaces. **C** $C_{PFS}$ values obtained by experiments and simulation of PMMA/ P(MMA-*sta*-PFS) films with various $X_c$. The hollow stars in (**C**) indicate the specific $X_c$ values, at which $C_{PFS} = 25\%$, chosen for the investigations of surface dynamics. $X_c$ was the weight fraction of copolymers in the PMMA matrix. Definition of experimental and simulation $C_{PFS}$ are shown in Supplementary Eq. 5 and Supplementary Eq. 4, respectively, in the SI. The experimental $C_{PFS}$ were deduced from the water contact angles shown in Supplementary Fig. 7. Error bars are ± standard errors.

concentration of PFS was kept constant ($C_{PFS}$ = 25%; see Fig. 1C; the rest of PFS monomers was buried beneath the surface, see Fig. 1A) and the loop size was changed by varying the $f_{PFS}$ of the copolymers. Simulation results showed that the average surface loop depth ($d_{loop}$) decreased from 10.3 to 4.9 nm with $f_{PFS}$ increasing from 0 to 0.11 (Fig. 1B), indicating that the chain loops extended deeper into the films interior at lower $f_{PFS}$.

Before discussing the effects of the loop size on the surface dynamics, three pieces of information are noteworthy. (1) $T_g$ of the P(MMA-*sta*-PFS) with $f_{PFS}$ in the range from 0.022 to 0.106 is quite close to that of PMMA (see Supplementary Table 1). This means that the introduction of P(MMA-*sta*-PFS) does not change the average dynamics of the PMMA films. (2) The similar reactivity ratios between MMA and PFS ensures the formation of a statistic copolymer with random distribution of PFS in the chains (see details in Section 2.3 in the SI). The random structure prevents the PFS units in the copolymer chains from aggregating, resulting in a uniform distribution of P(MMA-*sta*-PFS) within the blended films, as confirmed by small angle X-ray scattering (SAXS) (Supplementary Fig. 8). Such copolymer chain structure also guarantees that the relaxation of PFS is governed by the dynamics of the neighboring MMA. (3) Not any detectable phase separation was observed at the surface of the films with $C_{PFS}$ = 25%, (Supplementary Fig. 10). Taken all together, the surface enrichment of PFS would only facilitate the formation of chain loops at the surface without changing the dynamics; thus, any variations in surface dynamics are likely due to the presence of surface loops with controlled size.

## Surface mobility measured by experiments and simulations

Surface dynamics were assessed via surface loss tangent (tan$\delta$) measurements obtained using amplitude modulation-frequency modulation atomic force microscopy at a frequency of 330 kHz. Tan$\delta$ is related to the energy dissipation caused by internal friction due to segmental motion. As shown in Fig. 2, there is a significant increase in tan$\delta$ values and a broadening of the distribution with increasing temperature from 298 to 343 K for the PMMA films. The increase in tan$\delta$ values with increasing temperature indicates that the surface of the PMMA films is in the glassy or sub-$T_g$ state (see the definition in

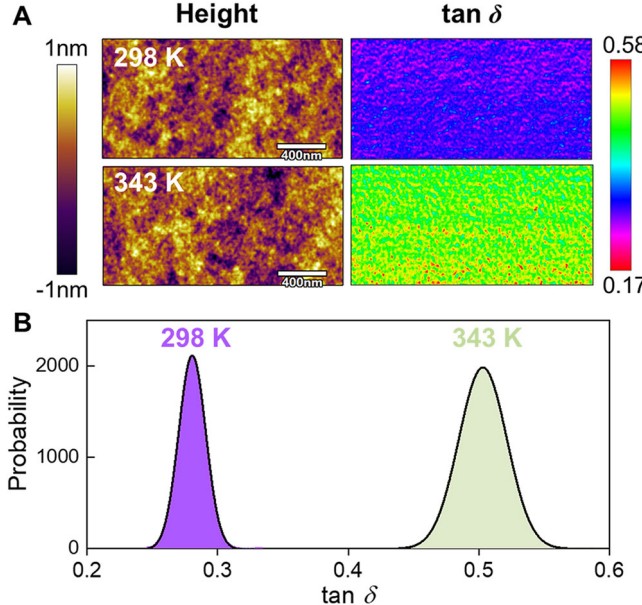

**Fig. 2 | Surface loss tangent and morphology of PMMA films. A** Surface topographic and loss tangent images of PMMA films at 298 and 343 K; **B** statistical distributions of surface tan$\delta$ values. See section 10 in the SI for the experimental details. ($h$ = 300 nm).

Supplementary Fig. 9), where thermal activation of the segmental motion increases the dissipation of energy. The broad distribution of tan$\delta$ values (Fig. 2) indicates the substantial dynamical heterogeneity at the surface, in which the soft spots with faster dynamics can dissipate more energy, whereas low tan$\delta$ values occur on harder surface areas with arrested dynamics[16,17,19,48]. Notably, compared to bulk PMMA, where tan$\delta$~0.1 in the glassy state (Supplementary Fig. 9), the higher tan$\delta$ values (i.e., tan$\delta$≈0.27 and 0.51 at 298 and 343 K, respectively) suggest an extraordinary enhancement in segmental mobility at the surface, or the outermost surface probed by AFM may already be in the glassy to the rubbery transition state. The increased surface mobility and energy dissipation of polymer glasses are consistent with the surface viscoelasticity measurements made by Kajiyama and coworkers in the late 1990s and early 2000s[48–50].

Supplementary Fig. 10 shows the tan$\delta$ images of PMMA/P(MMA-*sta*-PFS) films with $X_c$ = 0.25% and various values of $f_{PFS}$, and the corresponding statistical distributions are displayed in Fig. 3A, B. The peak values of tan$\delta$ and the width of the distribution reflect the average mobility and dynamical heterogeneity across the surface, respectively. As shown in Fig. 3C, the peak value at 298 and 343 K decreased progressively, and the full width at half maximum (FWHM) of the distribution narrowed with increasing $d_{loop}$, indicating the suppression of surface dynamics with the reduced population of fast-relaxing domains at the surfaces occupied by larger loops. This means that after extending the loops deeper into the bulk, the surface layer segments experienced lower mobilities and smaller internal friction at a temperature far below the bulk $T_g$ of PMMA. The slowing of polymer surface dynamics induced by the extension of surface loops in the bulk differs from the decoupling of surface and bulk dynamics observed in ordinary glasses[8,12,13,32]. This phenomenon suggests strong intramolecular coupling along the loops, which correlates the surface and bulk dynamics.

To further clarify how surface dynamics are influenced by chain loops, we employed MC simulations to calculate local segment mobilities near the surfaces. Our simulations were based on the coarse grain of the experimental polymers (see Section 6 in the SI) and combined the independent local vibrations of segments and a slithering snake algorithm of randomly generated kinks along chain molecules (Fig. 4B inset)[45,46]. This algorithm generates the collective motions of short-chain fragments along backbones, which are similar to the reptation motion of chain molecules in polymer melts[51]. Such a simulation model of intrachain sliding motion has been applied to predict the physical aging of ultrathin polymer glasses[45] and verified by recent experiments[52]. The instantaneous mobility in this model is defined according to Eq. 1[45,53].

$$\alpha(z, t) = 1 - \sum_{i=1}^{n} \sum_{j=1}^{N} \frac{\delta\left[z_{ij}(t) - z_{ij}(t - \Delta t)\right]}{N_{tot}(z, t)} \tag{1}$$

where $z_{ij}(t)$ represents the $z$-axis position of monomer $j$ on the $i$th chain at the $t$th MC step within the polymer thin films and $N_{tot}(z, t)$ denotes total number of monomers at the $z$ position at time $t$. The local mobility within the thin film along the $z$-axis is defined as the time average of $\alpha(z) = \langle \alpha(z, t) \rangle$. Figure 4A shows the total mobility distributions, including the contributions of local vibrations and intrachain sliding motions, near the free surface of films with distinct $d_{loop}$. A peak in mobility was evident near the 8-nm–thick polymer–air boundary region, indicating the presence of a mobile surface layer in which the segments moved faster than bulk segments. The surface mobilities of the films decreased with increasing distance from the free surfaces, forming a mobility gradient. Notably, as $d_{loop}$ increased, a weaker surface gradient with a lower magnitude of surface enhancement was observed; Fig. 4A. This finding was consistent with the experimental observations of reduced surface

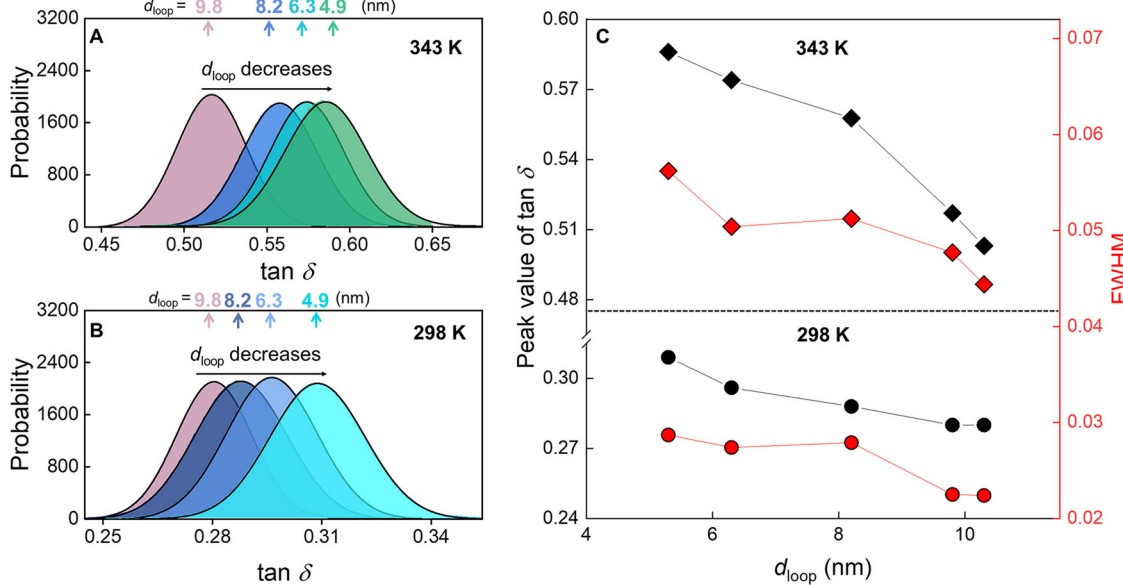

**Fig. 3 | Surface loss tangent of PMMA/P(MMA-*sta*-PFS) films with various $f_{PFS}$.** Distribution of tan$\delta$ values at **A** 343 K and **B** 298 K for PMMA/P(MMA-*sta*-PFS) films with various values of $f_{PFS}$ and $d_{loop}$. Films with $f_{PFS} > 0$ had the same $C_{PFS}$ of 25%. ($h = 300$ nm). **C** The FWHM of the distribution profiles in (**A**, **B**) for films with various $d_{loop}$ and at different temperatures.

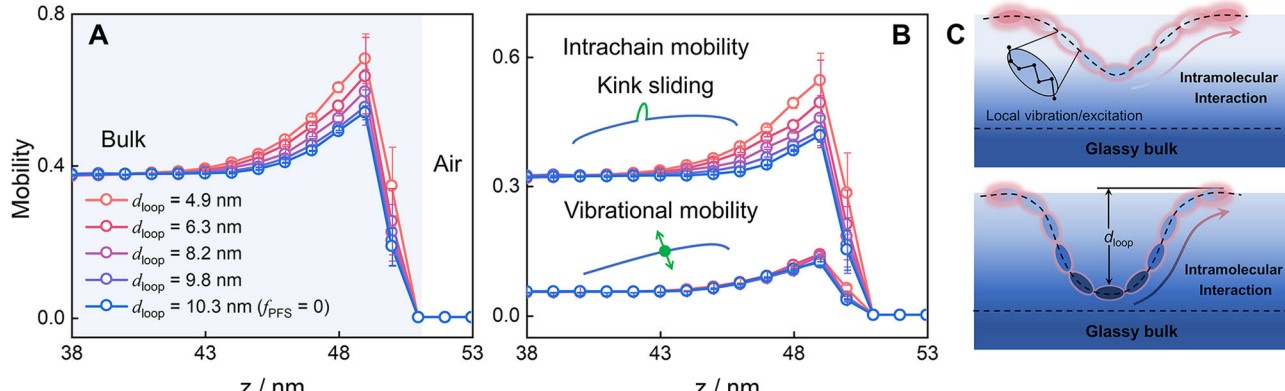

**Fig. 4 | Segmental mobility in the near surface region of simulated PMMA/ P(MMA-*sta*-PFS) films. A** Spatial distributions of segmental mobilities near the free surface of films with distinct $f_{PFS}$ and **B** the mobility gradients of local vibrational and intrachain mobilities near the surface. **C** Schematic illustrations of the variation in surface dynamics caused by intrachain coupling along loops. Insets in (**B**) display the sliding motions of kinks along chains and the vibrations of segments. Error bars are ± standard errors.

dynamics and lower surface dissipation for polymer films with a larger $d_{loop}$ (Fig. 3C).

Next, we computed the vibrational and intrachain mobilities individually to determine the effect of intramolecular dynamic coupling on surface relaxation. The local vibrations that were irrelevant to chain connectivity generated the same mobility gradients; Fig. 4B. This was in line with the simple molecular and atomic glasses in which the surface diffusion is decoupled from the bulk dynamics[8,12,13]. However, the intrachain mobility near the surface was more strongly constrained by the larger loops extending deeper into the dynamically sluggish bulk (Fig. 4B). Notably, a comparison of the surface gradient of total mobility (Fig. 4A) and its intrachain components (Fig. 4B) demonstrated that the variations in the intrachain mobilities caused by kink sliding contributed entirely to the change in total surface mobility for polymers with various $d_{loop}$ values, demonstrating the significant impact of intrachain coupling on surface dynamics.

Overall, the experimental and simulation results demonstrated that intramolecular dynamic interactions with the slow-relaxing segments on loops penetrating the film interior slowed the fast relaxation at surfaces. This phenomenon can be explained in the framework of the dynamics facilitation theory, which assumes that the localized active spots aid the relaxation of an adjacent region, and that dynamic facilitation propagates along strings to relax the system[54]. For polymers, intramolecular barriers for backbone reorientation, together with chain connectivity, increase coupling between *intrachain* relaxations, causing intramolecular dynamic facilitation[36]. Near surfaces, as illustrated in Fig. 4C, the surface contacts of chains with excess free volumes[41,42] act as effective local excitations, which facilitate a sequence of relaxations along loops, leading to fast relaxation of surface segments. In turn, slow relaxational units on loops deeper in the bulk retard the fast relaxation at the surface in the chain loops. Therefore, for films with loops extending deeper into the bulk, slower relaxing beads on the bottom of the loops would slow or even terminate this facilitation process, increasing the surface

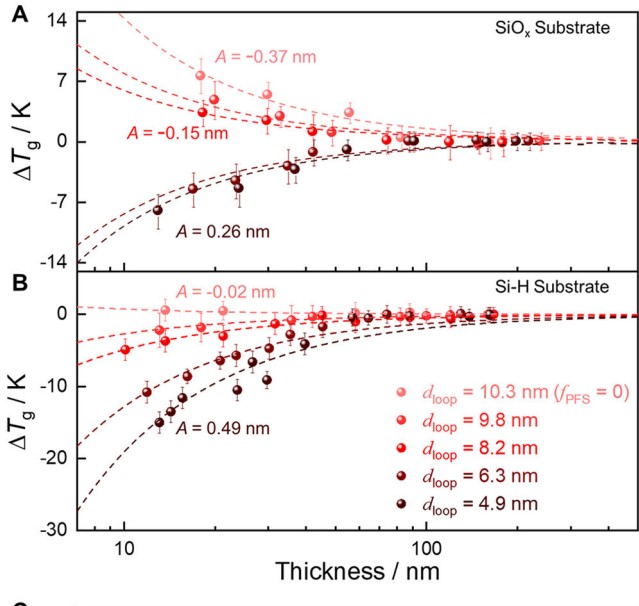

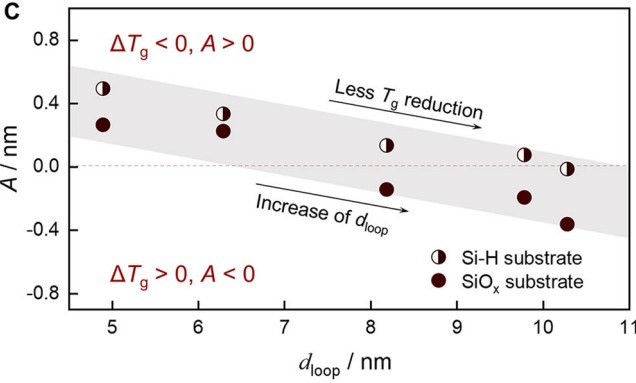

**Fig. 5 | Glass transition of thin films tailored by the size of surface loops.** $\Delta T_g$ of thin films on **A** SiO$_x$ and **B** Si–H substrates. **C** Variations in the $A$ value with $f_{PFS}$ and $d_{loop}$. The dashed curves in (**A**, **B**) are the fitting results using Eq. 2 with a fixed $\delta$ of 1. Positive/negative $A$ values are associated with a decreasing/increasing trend of $T_g$ upon confinement. Standard errors of the $A$ values were included in Supplementary Table 4 in the SI. Error bars are ± standard errors.

activation energy and decreasing the magnitude of surface gradients. This scenario is consistent with the sliding concept in thin films[41,42], in which surface loops effectively plasticize the material by kink transmission along chains, whereas the segments in the slow bulk provide bottlenecks that decelerate the movement of kinks.

### Tailoring the properties of polymer thin films

The suppression of surface dynamics by intramolecular coupling can be applied to design a stable polymer film with a higher $T_g$ by optimizing the surface chain conformation, *i.e.*, loop depth. The thermal $T_g$ of ultrathin PMMA/P(MMA-*sta*-PFS) films with $C_{PFS} = 25\%$ and various $d_{loop}$ values on native oxide layer-covered Si (SiO$_x$) and hydrogen-passivated silicon (Si–H) was measured using temperature-variable ellipsometry (see section 13 in SI for experimental details). On both substrates, a smaller $T_g$ depression relative to the bulk value was observed for films with longer loops; Fig. 5A, B. The thickness dependence ($h$) of thermal $T_g$ was described using the relationship proposed by Keddie et al.[1]:

$$\Delta T_g = T_g(h) - Tg, \text{bulk} = -Tg, \text{bulk}(A/h)^\delta \qquad (2)$$

Here, the exponent $\delta$ is set to 1. Fitting each $\Delta T_g - h$ curve with Eq. 2 produces an $A$ value, which quantifies the effect of thin film

confinement on the $T_g$. Larger $A$ values are associated with a more substantial reduction in the $T_g$ and vice versa; Fig. 5A, B. The $A$ values of films on SiO$_x$ were lower than those of films on Si–H because of the favorable hydrogen-bonding interactions between PMMA and SiO$_x$; Fig. 5C. However, the $A$ values of films on both substrates decreased with $d_{loop}$. Because the decrease in the $T_g$ is caused by dynamical enhancement at the free surface[1,5,55–57], the lower $A$ values and less reduction in the thermal $T_g$ for films with a larger $d_{loop}$ suggest that polymer surfaces occupied by chain loops extending deeper into the film interior have slower dynamics than those with shorter loops. This result further highlights that intramolecular coupling can affect the dynamics of polymers near surfaces, providing new insights into engineering polymeric devices at the nanoscale where the interface dominates the properties.

## Discussion

We demonstrated that the accelerated relaxation of surface polymers was suppressed by slow bulk polymers because of the strong intramolecular coupling associated with chain connectivity along the surface chain loops. Such effect causes that the polymer films having surface chain loops penetrating deeper into the bulk are associated with a less mobile, thinner surface layer with reduced magnitude of dynamical gradients. The finding highlights the important role of intramolecular dynamical correlations on the surface dynamics of polymer glasses, and on the other hand, also reinforce the concept that molecular relaxations at surfaces can be affected by the extent of the molecules coupled to the bulk[58,59]. Thus, our findings enable new insights into the nature of the surface of macromolecular glasses and also allow the opportunity to tailor polymer dynamics at interfaces and under nanoscale confinement by optimizing the arrangement of polymer chains at the surface and manipulating the bulk-to-surface dynamic interactions arising from intrachain coupling.

Based on the findings that the intramolecular interaction can propagate slow dynamics along the chain loops, we could expect that at a substrate interface, the adsorption of segments could suppress the mobility of the ones in the loops that extend far from the interfaces. Particularly, when the film is sufficiently thin, the slow dynamics due to interfacial adsorption can surpass the free surface effect that accelerates polymer dynamics. Thus, our results could provide additional insights into understanding the coupling and intertwining of interfacial and free surface effects in thin polymer films and the related phenomena, such as interfacial adsorption erasing the surface effect, causing the dynamics of thin films to recover to the bulk state[60–62].

## Methods

### Experiments

The P(MMA-*sta*-PFS) copolymers and PMMA homopolymer with molecular weights around 86-87 kg/mol were synthesized through atom transfer radical polymerization. The $f_{PFS}$ of the copolymers was tuned between 2.2 and 10.6% by changing the feeding ratio of MMA and PFS. PMMA/P(MMA-*sta*-PFS) films were fabricated using a solution blend method: P(MMA-*sta*-PFS) and PMMA were co-dissolved in trifluorotoluene at prescribed molar ratios and subsequently spin-casted on substrates. The blended films were then annealed above $T_g$ ($T_g + 35$ K) to expedite the surface segregation of the PFS and drive the formation of surface loops. The copolymer fraction in the blended films was carefully tuned to generate a constant surface concentration of PFS, $C_{PFS} = 25\%$; and the surface loss tangent and the $T_g$ of the blended films were measured by amplitude modulation-frequency modulation AFM and ellipsometry, respectively. Experimental details are included in SI.

### Coarse-grained simulations

The P(MMA-*sta*-PFS) with distinct $f_{PFS}$ has molecular weight around 86-87 kg/mol, which corresponds to the total coarse-grained monomer of around $N \approx 200$ by choosing 4 repeat units of PMMA and PFS as a

monomer with size of 1.0 nm, see details of mapping in SI. The number of PFS monomer takes the value of 4, 10, 16, and 22, yielding the mole fractions of $f_{PFS}$ in simulations with 0.02, 0.05, 0.08, 0.11, which are consistent with experimental parameters shown in Supplementary Table 1. We used the lattice model of polymer chains to study the mobility of P(MMA-*sta*-PFS) thin films. The motion of polymer chains is generated through a micro-relaxation model, which allows each segment to change positions with its neighboring solvent sites, accompanied by sliding diffusion along the chain direction if necessary. Conventional metropolis sampling was employed in each micro-relaxation step with the potential energy change:

$$\frac{\Delta E}{k_B T} = \left( b_0 \frac{B_{mf}}{E_c} + b_1 \frac{B_{ma}}{E_c} + b_2 \frac{B_2}{E_c} + b_3 \frac{B_3}{E_c} + c \right) \frac{E_c}{k_B T} \quad (3)$$

Here $E_c$ is the bending energy for two adjacent bonds connected along the chain and $c$ is the total number of non-collinear connected bond pairs along the chain. $k_B$ is the Boltzmann constant and $T$ is the temperature. This model combines the kink-jump methods and the slithering diffusion terminated by extending the nearest kink conformation along the chain, originally proposed by de Gennes, which was proven to be a powerful technique to study the physical aging and surface crystallization of polymer thin films and solutions. Details of parameters and simulation setup to obtain the surface PFS fraction $C_{PFS}$ are summarized in SI.

### Simulation details
Methodological details of the simulations can be found in Section 6 of the SI.

## Data availability
The data that support the findings of this study are available within the article and its Supplementary Materials. All data are available from the corresponding author upon request.

## Code availability
The codes that support the findings of this study are available from Q.T. upon request.

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

## Acknowledgements

We acknowledge financial support from the National Natural Science Foundation of China (Grant nos. 22122306, 52373025, 22303084, and 12374207). Q.T. acknowledges the financial support by the open research fund of the Key Laboratory of Quantum Materials and Devices (Southeast University), Ministry of Education. The supercomputing resources at Beijing Super Cloud Computing Center (BSCC) are acknowledged. R.D.P. acknowledges support by the National Science Foundation (NSF) Materials Research Science and Engineering Center Program through the Princeton Center for Complex Materials (PCCM) (DMR-2011750).

## Author contributions

B.Z., Q.T., and R.D.P. conceived and supervised the experiments; H.T., J.L., and H.Z. performed experiments; Q.T. performed and analyzed the simulations; W.H. provided constructive advices for optimizing the simulation method and improvement on the article quality; all authors discussed results and wrote the manuscript.

## Competing interests

The authors declare no competing interests.
