## [Peer Review File · Nature Communications]

Intramolecular Dynamic Coupling Slows Surface Relaxation of Polymer GlassesReviewers' Comments:

Reviewer #1:

Remarks to the Author:

Tian et al. studied the surface dynamics in polymer films through a combination of experiments and simulations. They found a fascinating phenomenon-sluggish dynamics just beneath the top surface. This reviewer strongly recommends this study to be published in Nature Communications. Please provide clarification on the following points.

1. It seems that the experimental validation of the presence of loop chains at the surface is insufficient. The authors assume that P(MMA-sta-PFS) chains behave like a "Gibbs monolayer" in their experiments. Is this assumption correct? If, in reality, there is a concentration profile resembling surface enrichment observed in typical polymer blends, the discussion may take a different direction. First, could you clarify how you obtained the raw data for XPS? For instance, what was the emission angle for photoelectrons? Once you have this value, it will help establish the analytical depth. There is no guarantee that within this depth range, there exists a single P(MMA-sta-PFS) chain along the direction normal to the surface. Additionally, SFG signals are affected not only by the concentration of functional groups but also by their orientation. This implies that the interpretation is not straightforward for what the authors intend to claim. The analysis of contact angles using Cassie's equation also involves similar challenges. When the chain's morphology is not as simple as illustrated in panel A of Figure S3, for instance, where only one PFS unit within a single chain appears on the surface, and the rest of PFS units are submerged into the bulk, there can be additional complexities. Conversely, PFS moieties from other chains existing in a slightly deeper region may localize on the surface. If such structures exist, even if loop structures are present, their interpretation may vary in terms of length and other characteristics.

2. If my understanding is correct, the value of $\tan \delta$ in the glassy state should be much smaller than 0.1. The frequency they adopted was 330 kHz, so a similar statement should apply. In that case, the data from Figure 2 indicates that the surface is in a transition state or a rubbery state. This may not align with the authors' claim. Measuring $\tan \delta$ as a function of temperature for bulk samples would be beneficial to clarify this discussion.

Reviewer #2:

Remarks to the Author:

The manuscript of Tian et al is a very interesting report on the origin of surface relaxation in glasses. The topic is of interest to a large number of researcher crossing several field (physics, chemistry, engineering, ...). By means of experiments and simulations, the authors verify that surface dynamics can be tuned by adequately modifying the chemistry fo the system. While the dynamics of polymer glasses is considered to be faster than in bulk, the authors show that an increase in intermolecular coupling --achieved by extending chain loops into the film-- leads a reduction of the 'free surface effect'. The results are unprecedented and could be of interest to a large number of researchers, including both experimentalists and theorists.

A revised version of the manuscript could be considered for publication in Nature Communications.

Main remark

The authors use $\tan \delta$ ($=E''/E'$) to assess changes in surface dynamics. I agree with their interpretation, an increase in $\tan \delta$ indicates a region with larger dissipation, but this does not univocally correlates to faster dynamics as they claim. In the frequency domain, $\tan \delta$ expresses --under rough approximation-- the derivative of a relaxation rate, and not the rate itself. By measuring $\tan \delta$ over a broad frequency range one would see a peak, centered around the main rate of a molecular relaxation mode; as $\tan \delta$ goes to zero both when probing the material at higher and at lower rates than the frequency of the peak, one cannot say if dynamics speeds up or slows down by

looking at a change in τ , especially if the position of the peak maximum is not known. As a consequence it is not that straightforward to interpret a measurement of τ at constant excitation frequency, as done in this manuscript. The authors should better clarify these ideas and better present the physical framework they used to interpret their experimental results.

Minor points

a) The mechanism described by the authors to rationalize the coupling between surface relaxation and intermolecular coupling is particularly interesting. What would happen in case of an adsorbing interface? Would the propagation of loops from the adsorbed layer be able to reduce the free surface effect on thermal T_g (see ACS Macro Lett. 2017, 6, 354–358) and on segmental dynamics (see PRL 119, 097801 (2017))?

b) While the text is usually accessible to general readers, some paragraphs and the abstract, could be improved by simplifying the wording and avoiding technical terms. For example, at page 2, we read about "the bulk fictive temperature", a quantity which is not particularly relevant in the discussion and was not properly introduced.

c) The authors should not confuse thermal T_g and dynamics. While the two could be coupled in bulk, they might be significantly different in confinement, where a significant shift in thermal T_g is accompanied by minor or null changes in segmental dynamics.

d) When referring to statistical distribution, see Fig 2B, we should read probability. The typo should be corrected all over the text.

Reviewer #3:

Remarks to the Author:

Tian, et al. Intramolecular Dynamic Coupling Slows Surface Relaxation of Polymer Glasses

This manuscript is interesting, using a cleverly designed system to quantify the effect of larger loop length on surface dynamics of polymer glass-formers. The authors do a good job of convincing the reader that they see such an effect in their data. There are just a few points that require clarification prior to publication.

1) English could use some work in spots, such as "This leads to that"

2) The figure 1c x-axis label is Weight fraction of copolymer (X_c). This reviewer worries that the units of the numbers on the x-axis might be % instead of fraction. Is the largest X_c data at fraction 0.67 or 0.67 wt%?

3) P. 4 "three pieces of information are noteworthy" has three numbered items following that are stated as though they were facts and sometimes it is not so easy for the reader to understand whether they are facts or speculations by the authors. For instance, T_g being the same as PMMA might be interpreted that there are small domains of PFS embedded in the PMMA matrix, forming flower micelles, as that morphology would not change the PMMA matrix T_g . This reviewer doubts those would affect any of the main results of this manuscript but probably this possibility should be discussed? Also, "prevented the formation of dimers, trimers and larger PFS sequences" is not at all obvious. Anything that is unproven speculation should have a caveat, such as being preceded by We hypothesize.

4) What fraction of the PFS monomers are at the surface? Where are the rest of them?

5) Figure 5c: What is the uncertainty in A?

Reviewer #1 (Remarks to the Author):

Tian et al. studied the surface dynamics in polymer films through a combination of experiments and simulations. They found a fascinating phenomenon-sluggish dynamics just beneath the top surface. This reviewer strongly recommends this study to be published in Nature Communications. Please provide clarification on the following points.

Responses: We thank the reviewer for the encouraging comments. In the following, we have provided point-by-point replies to clarify the main concerns raised in your comments.

Comment 1:

It seems that the experimental validation of the presence of loop chains at the surface is insufficient. The authors assume that P(MMA-sta-PFS) chains behave like a “Gibbs monolayer” in their experiments. Is this assumption correct? If, in reality, there is a concentration profile resembling surface enrichment observed in typical polymer blends, the discussion may take a different direction. First, could you clarify how you obtained the raw data for XPS? For instance, what was the emission angle for photoelectrons? Once you have this value, it will help establish the analytical depth. There is no guarantee that within this depth range, there exists a single P(MMA-sta-PFS) chain along the direction normal to the surface.

Responses: We thank the reviewer for your insightful comments. The “Gibbs monolayer” assumption may not be a good illustration. The loop chain picture is constructed from Monte Carlo (MC) simulation data based on the coarse-grained models of the **same experimental systems**, where the loops are statistically formed near surfaces due to the surface enrichment of PFS monomers. This surface enrichment of PFS is also supported by experimental data. We agree that there may be a gradient of PFS concentration near the surface, which could induce formation of some chain tails, as illustrated in Fig. S3A. However, localization of a large amount of PFS at outmost surface induce formation of considerable fraction of surface loops. This was also verified by our MC simulation.

The takeoff angle of X-ray in the XPS measurement is 15°, corresponding to a detection depth of ~ 2.3 nm. The sampling depth is smaller than the penetration depth of surface loops (Fig. 1B), and thus the surface F/C ratio revealed the concentration of PFS in the outmost surface layer of the films. Adsorption of the PFS at the outmost surface drives the formation of surface loops.

According to the reviewer advices, we added a paragraph in page S6 to describe the experimental procedures of SFG, contact angle and XPS. The takeoff angle and sampling depth for XPS measurements were incorporated in this paragraph and in the caption of Fig. S3 (“**The takeoff angle of photoelectrons in the XPS measurement is 15°, corresponding to a sampling depth of 2.3 nm.**”).

Additionally, SFG signals are affected not only by the concentration of functional groups but also by their orientation. This implies that the interpretation is not straightforward for what the authors intend to claim. The analysis of contact angles using Cassie’s equation also involves similar challenges.

Responses: Although the SFG spectra and water contact angle were affected by both the concentration and orientation of PFS at surface, the consistency of the results with compositional analysis by XPS indicates that surface segregation of PFS and depletion of MMA are the main factors affecting the SFG signal and contact angles on the surface of PMMA/P(MMA-*sta*-PFS) blend films. As shown in Fig. S3, we observed a decay of the MMA signal in the SFG spectra and concurrent increase in the water contact angle and surface F/C ratio. The full consistence among results from these methods indicate that the changes in water contact angle and SFG signal from MMA are associated with the surface segregation of PFS and depletion of MMA, rather than the changes of orientation.

We added “Although the SFG spectra and water contact angle were affected by both the concentration and orientation of PFS at surface, the consistency of the results with compositional analysis by XPS indicates the negligible influence of orientation changes and evidences surface segregation of PFS and depletion of MMA from the surface during annealing” in the caption of Fig. S3 to make the explanation of SFG and contact angle data clearer.

When the chain's morphology is not as simple as illustrated in panel A of Figure S3, for instance, where only one PFS unit within a single chain appears on the surface, and the rest of PFS units are submerged into the bulk, there can be additional complexities. Conversely, PFS moieties from other chains existing in a slightly deeper region may localize on the surface. If such structures exist, even if loop structures are present, their interpretation may vary in terms of length and other characteristics.

Responses: We agree with the reviewer that the P(MMA-*sta*-PFS) chains would not form a simple structure like the Gibbs monolayer at the vapor interface, as shown in Fig. S3A. We used Fig. S3A as an ideal schematic illustration of the formation of surface loops due to surface segregation of the low-surface-energy units of the PFS. As pointed out by this reviewer, some of the PFS units may be submerged into the film interior, while the rest were anchored to the surface, and occasionally some chain tails may exist at surface. To clarify the fraction of surface chains with PFS forming loops, we performed dynamic Monte Carlo (MC) simulations based on the coarse-grained models of **the same experimental systems**. After equilibration of the systems at high temperature to have the PFS to be enriched at the surface, MC showed that more than 85% surface chains take loop conformation (Table R1 and Table S3). Therefore, the loop formation dominates the conformation of surface chains with PFS monomers. We also plot a typical loop formation on the film surface from simulations, see snapshot Inset of Fig. 1B.

Table R1. The fraction of surface chains forming loops at the surface

f_{PFS}	fraction of surface chains forming loops (%)
0.02	85.1
0.05	86.1
0.08	87.7
0.0105	85.9

For the second scenario proposed by reviewer, we calculated the probability of a PFS located in a subsurface region within 4 nm from vapor interface, but is not belong to the polymers with other PFS at the out surface, these probabilities calculated from our simulations are less than 5%, therefore we believe that such effect could be negligible.

To avoid the potential misinterpretation, we modify Fig. S3A to be consistent with the realistic picture, in which some PFS of the surface chain may be buried in the film interior and there also exist some chain tails.

Table R1 is added to SM (Section 7) as Table S3 to illustrate that the majority of surface chains forms loops.

Also, we add sentences in page 4 to emphasize the support from simulation and note the existence of both loops and tails at surface.

"Figures 1A and Inset of Figure 1B show the distribution of PFS near the surface region and a snapshot of a simulated surface P(MMA-sta-PFS) chains. More than 85% surface chains with PFS located on the surface form loops (Table S3). Inset of Figure 1B depicts typical Gaussian-like surface loops formed by polymer strands between the surface-anchoring PFS units. Statistically, the surface loops penetrated deeper into the films mixed with copolymers with larger f_{PFS} (Figure 1B)."

Comment 2:

If my understanding is correct, the value of $\tan \delta$ in the glassy state should be much smaller than 0.1. The frequency they adopted was 330 kHz, so a similar statement should apply. In that case, the data from Figure 2 indicates that the surface is in a transition state or a rubbery state. This may not align with the authors' claim. Measuring $\tan \delta$ as a function of temperature for bulk samples would be beneficial to clarify this discussion.

Responses: We thank the reviewer for the constructive comments. It makes it possible to get a clearer picture of the dynamic state at the surface of the glassy PMMA films by addressing this comment.

According to your advices, we performed temperature-sweep dynamical mechanical analysis (DMA) test on bulk PMMA at frequencies of 1 ~ 20 Hz; Fig. R1(A). Apparently, the

$\tan\delta$ value of PMMA in the glassy state is about 0.1 and this value increases to 0.7 at T_g and then decays to ~ 0.1 at the rubbery state (Fig. R1). The larger $\tan\delta$ values (*i.e.*, $\tan\delta = 0.27$ and 0.51 at 298 and 343 K, respectively; Fig. 2) at surface than the value in the glassy bulk suggests an extraordinarily enhanced segmental mobility at the surface, or the outermost surface layer probed by AFM may already be in the glassy to rubbery transition state. As well, the fact of increase of surface $\tan\delta$ from 0.27 to 0.51 with increasing temperature from 298 to 343 K (Fig. 2) also indicates that the surface of glassy PMMA films is in the sub- T_g states; see Fig. R1.

We revised the statements on dynamic state at surface of PMMA films by describing it as having enhanced mobility or in the glassy-to-rubbery transition state; see the revisions in the paragraph crossing page 5 and 6; [Notably, compared to bulk PMMA, where $\tan\delta \sim 0.1$ in the glassy state (Figure S8), the higher $\tan\delta$ values (*i.e.*, $\tan\delta \approx 0.27$ and 0.51 at 298 and 343 K, respectively) suggest an extraordinary enhancement in segmental mobility at the surface, or the outermost surface probed by AFM may already be in the glassy to rubbery transition state. The increased surface mobility and energy dissipation of polymer glasses is consistent with the surface viscoelasticity measurements made by Kajiyama and coworkers in the late 1990s and early 2000s].

Meanwhile, Fig. R1 is included in the SM as Fig. S8 to help clarify this point.

Figure R1. (A) the $\tan\delta \sim T$ curve of PMMA measured by DMA at different frequencies. (B) the 3D plot of $\tan\delta$ as a function of temperature and frequency; the red circles represent the measured $\tan\delta$ values (*i.e.*, curves in panel A). The 3D plot in panel B was constructed by extrapolating the measured $\tan\delta \sim$ temperature curves to wider frequency ranges on the basis of the VFT relation between frequency and temperature (*Phys. Rev. B* 2003, 67, 174202). The shape of $\tan\delta \sim T$ curve was assumed invariant with the frequencies.

Reviewer #2 (Remarks to the Author):

The manuscript of Tian et al is a very interesting report on the origin of surface relaxation in glasses. The topic is of interest to a large number of researcher crossing several field (physics, chemistry, engineering, ...). By means of experiments and simulations, the authors verify that surface dynamics can be tuned by adequately modifying the chemistry for the system. While the dynamics of polymer glasses is considered to be faster than in bulk, the authors show that an increase in intermolecular coupling --achieved by extending chain loops into the film-- leads a reduction of the 'free surface effect'. The results are unprecedented and could be of interest to a large number of researchers, including both experimentalists and theorists.

A revised version of the manuscript could be considered for publication in Nature Communications.

Responses: We thank the reviewer for the positive comments and appreciations to our work.

Comment 1:

Main remark

The authors use $\tan\delta$ ($= E'/E''$) to assess changes in surface dynamics. I agree with their interpretation, an increase in $\tan\delta$ indicates a region with larger dissipation, but this does not univocally correlates to faster dynamics as they claim. In the frequency domain, $\tan\delta$ expresses -- under rough approximation -- the derivative of a relaxation rate, and not the rate itself. By measuring $\tan\delta$ over a broad frequency range one would see a peak, centered around the main rate of a molecular relaxation mode; as $\tan\delta$ goes to zero both when probing the material at higher and at lower rates than the frequency of the peak, one cannot say if dynamics speeds up or slows down by looking at a change in $\tan\delta$, especially if the position of the peak maximum is not known. As a consequence is it no that straightforward to interpret a measurement of $\tan\delta$ at constant excitation frequency, as done in this manuscript. The authors should better clarify these ideas and better present the physical framework they used to interpret their experimental results.

Responses: We thank the reviewer for the constructive comments which help improve our work. By collecting the $\tan\delta \sim$ temperature relations of bulk PMMA, which allow the estimation of the peak temperature of $\tan\delta$ at the surface, we have confirmed that there is a positive correlation between the surface loss tangent and the surface dynamics.

Fig. R1(B) shows the $\tan\delta \sim T$ curves obtained by DMA at various frequencies and the 3D spectrum across a wide range of frequencies and temperatures derived from the experimental $\tan\delta \sim T$ curves and VFT time-temperature relations for bulk PMMA (Phys. Rev. B 2003, 67, 174202). The bulk T_g of PMMA at frequency of 330 kHz (*i.e.*, the frequency of surface dissipation measurement by AFM) is as high as 452 K. As the surface T_g is approximately 40 ~ 60 K lower than bulk T_g (Polymer 1998, 39, 4665), it is reasonable to infer that the T_g at surface of PMMA at measurement frequency of 330 kHz cannot be less than 350 K. Accordingly, the surface of PMMA at 298 and 343 K are in the glassy or sub- T_g region, where the $\tan\delta$ increases with temperature as well as the segmental mobility. Moreover, an increase of surface $\tan\delta$ from 0.27 to 0.51 with increasing the temperature (*i.e.*, increasing segmental mobility) from 298 to 343 K, as shown in Fig. 2, further

confirmed the positive correlations between $\tan\delta$ and the surface mobility.

In the revised manuscript, a brief discussion of the relationships between $\tan\delta$ and surface mobility is included in the paragraph across pages 5 and 6, and the detailed reasoning is instead included in Section 11 of the SM.

Comment 2:

Minor points

a) The mechanism described by the authors to rationalize the coupling between surface relaxation and intermolecular coupling is particularly interesting. What would happen in case of an adsorbing interface? Would the propagation of loops from the adsorbed layer be able to reduce the free surface effect on thermal T_g (see ACS Macro Lett. 2017, 6, 354–358) and on segmental dynamics (see PRL 119, 097801 (2017))?

Responses: Thank the reviewer for the reflections. Our results demonstrated that the intramolecular interaction can propagate slow dynamics along the interfacial loops. Based on this finding, we anticipate that, at adsorbing interface, the adsorption of segments on the substrate interface could slow the dynamics of segments in loops extending far away from the interface, due to the intrachain dynamical couplings. Particularly, when the film is sufficiently thin, the slow dynamics due to interfacial adsorption can surpass the free surface effect that accelerates polymer dynamics and reduce the free surface effect. In this aspect, our results provide rational explanation on the phenomena of dynamic interactions between interface and free surface effects in thin polymer films, as reported in ACS Macro Lett. 2017, 6, 354 and Phys. Rev. Lett. 2017, 119, 097801, that the interfacial adsorption "erases" surface effect, causing recovery of dynamics of thin films to the bulk state.

We have included the following discussion on the role of intramolecular interaction at a substrate interface in the last paragraph of the revised manuscript;

["Based on the findings that the intramolecular interaction can propagate slow dynamics along the chain loops, we could expect that at a substrate interface, adsorption of segments could suppress mobility of the ones in the loops that extend far from the interfaces. Particularly, when the film is sufficiently thin, the slow dynamics due to interfacial adsorption can surpass the free surface effect that accelerates polymer dynamics. Thus, our results could provide additional insights into understanding the coupling and intertwining of interfacial and free surface effects in thin polymer films and the related phenomena, such as interfacial adsorption "erasing" the surface effect, causing the dynamics of thin films to recover to the bulk state [60, 61]"].

Comment 3:

b) While the text is usually accessible to general readers, some paragraphs and the abstract, could be improved by simplifying the wording and avoiding technical terms. For example, at page 2, we read about "the bulk fictive temperature", a quantity which is not particularly relevant in the discussion and was not properly introduced.

Responses: We checked carefully language and expression of the manuscript.

The sentence,

"Zhang and Fakhraai disclosed that the surface diffusion coefficients of molecular glasses of N,N'-bis(3-methylphenyl)-N,N'-diphenylbenzidine were almost unchanged when the bulk fictive temperature was varied over a range of 35 K, equivalent to 13 to 20 orders of magnitude in bulk relaxation times"

is modified as

"Zhang and Fakhraai disclosed that the surface diffusion coefficients of ordinary and ultra-stable molecular glasses of N,N'-bis(3-methylphenyl)-N,N'-diphenylbenzidine are almost undistinguished although the bulk relaxation times differ by about 13 to 20 orders of magnitude" (page 2).

Comment 4:

c) The authors should not confuse thermal T_g and dynamics. While the two could be coupled in bulk, they might be significantly different in confinement, where a significant shift in thermal T_g is accompanied by minor or null changes in segmental dynamics.

Responses: We thank the reviewer for your nice suggestions. To be precise, we measured the thermal T_g of thin films by tracking the temperature dependence of thermal expansion of films using ellipsometry, Fig. S10.

We changed " T_g " to "thermal T_g " throughout the **Section C** in the **Results** part for clarity and accuracy.

Comment 5:

d) When referring to statistical distribution, see Fig 2B, we should read probability. The typo should be corrected all over the text.

Responses: Thank the reviewer for the kindly reminding, **we have corrected the spelling mistake all over the text.**

Reviewer #3 (Remarks to the Author):

Tian, et al. Intramolecular Dynamic Coupling Slows Surface Relaxation of Polymer Glasses

This manuscript is interesting, using a cleverly designed system to quantify the effect of larger loop length on surface dynamics of polymer glass-formers. The authors do a good job of convincing the reader that they see such an effect in their data. There are just a few points that require clarification prior to publication.

Responses: We thank the reviewer for your encouraging and insightful comments on our manuscript.

Comment 1:

1) English could use some work in spots, such as “This leads to that”

Responses: We have carefully reviewed and polished the English of the manuscript. "This leads to that the accelerated segmental relaxation on polymer glass surfaces markedly slows when the surface polymers extend chain loops deeper into the film interior." is changed to " **Such effect leads to that accelerated segmental relaxation on polymer glass surfaces markedly slows when the surface polymers extend chain loops deeper into the film interior**" (page 1).

Comment 2:

2) The figure 1c x-axis label is Weight fraction of copolymer (X_c). This reviewer worries that the units of the numbers on the x-axis might be % instead of fraction. Is the largest X_c data at fraction 0.67 or 0.67 wt%?

Responses: The unit of the numbers on the x-axis is weight fraction of copolymer (X_c) in the blends. The largest X_c data is 0.67, corresponding to 67 wt% of copolymers in the blend films of PMMA/P(MMA-*sta*-PFS). Even at this large X_c , the fraction of PFS in the film is no more than 7.1%, which was supposed to exert negligible influence on the dynamics in the films.

We added a sentence " **X_c was the weight fraction of copolymers in the PMMA matrix**" in the caption of Fig. 1 to make it easy for readers to get the information.

Comment 3:

3) P. 4 “three pieces of information are noteworthy” has three numbered items following that are stated as though they were facts and sometimes it is not so easy for the reader to understand whether they are facts or speculations by the authors.

Responses: We thank the reviewer for the very helpful comments. Indeed, all the information was deduced from our experimental results. In the following, we explained how each piece of information was obtained.

(Item #1) As shown in Table S1, the T_g of the P(MMA-*sta*-PFS) with f_{PFS} in the range from 0.022 to 0.104 is quite close to that of PMMA. This result means that blend of P(MMA-*sta*-PFS) into PMMA matrix does not change the average dynamics of PMMA.

(Item #2) The similar reactivity ratios between MMA and PFS ensures the formation of a statistic copolymer with random distribution of PFS in the chains (see details in Section 2.3

in the SM). Results from small angle X-ray scattering (SAXS) (Fig. S7) measurement revealed that not any aggregation of PFS occurred in the blend films, further confirming that the PFS units was randomly distributed among the along the chain, without formation of larger PFS sequences.

(Item #3) From Fig. S9, we can see that not any detectable phase separation was observed at surface of the films with $C_{PFS} = 25\%$.

The above results and discussion were included in the first paragraph on page 5. We hope this can make the interpretation clear and straightforward.

[Before discussing the effects of the loop size on the surface dynamics, three pieces of information are noteworthy. (1) T_g of the P(MMA-*sta*-PFS) with f_{PFS} in the range from 0.022 to 0.106 is quite close to that of PMMA (see Table S1). This means that introduction of P(MMA-*sta*-PFS) does not change average dynamics of the PMMA films. (2) The similar reactivity ratios between MMA and PFS ensures the formation of a statistic copolymer with random distribution of PFS in the chains (see details in Section 2.3 in the SM). The random structure prevents the PFS units in the copolymer chains from aggregating, resulting in a uniform distribution of P(MMA-*sta*-PFS) within the blended films, as confirmed by small angle X-ray scattering (SAXS) (Figure S7). Such copolymer chain structure also guarantees that the relaxation of PFS is governed by the dynamics of the neighboring MMA. (3) Not any detectable phase separation was observed at surface of the films with $C_{PFS} = 25\%$, (Figure S9).]

For instance, T_g being the same as PMMA might be interpreted that there are small domains of PFS embedded in the PMMA matrix, forming flower micelles, as that morphology would not change the PMMA matrix T_g . This reviewer doubts those would affect any of the main results of this manuscript but probably this possibility should be discussed?

Responses: To examine if flower micelles or other kind of phase separation occur in the blend films, we conducted SAXS test on PMMA/P(MMA-*sta*-PFS) blends. As is clear in Fig. R2, the intensity smoothly decays with q for both PMMA and PMMA/P(MMA-*sta*-PFS). This indicates that there are no aggregation or phase separation in the blend films, and so the effect of phase separation can be excluded. The SAXS data was added in Section 9 of SM (Fig. S7) and the relevant discussion is also incorporated into the first paragraph on page 5 in the revised manuscript; [The random structure prevents the PFS units in the copolymer chains from aggregating, resulting in a uniform distribution of P(MMA-*sta*-PFS) within the blended films, as confirmed by small angle X-ray scattering (SAXS) (Figure S7)].

Figure R2. The SAXS profile of PMMA and PMMA/P(MMA-sta-PFS) blends with various f_{PFS} and X_c . The measurement was performed at ambient temperature after the samples equilibrated at 150 °C under vacuum for 24 h.

Also, “prevented the formation of dimers, trimers and larger PFS sequences” is not at all obvious. Anything that is unproven speculation should have a caveat, such as being preceded by We hypothesize.

Responses: We thank you for this nice suggestion. The reactivity ratios for copolymerization of MMA and PFS ($r_{\text{MMA}} = 0.88$ and $r_{\text{PFS}} = 1.17$; $r_{\text{MMA}} \times r_{\text{PFS}} = 1.03$; see Section 2.3 in the SM) and SAXS results (Fig. S7) of the blends suggest that the copolymer of P(MMA-sta-PFS) has a statistically random structure without large PFS sequences in the chains.

To make the statement precise and clear, old sentence on page 5:

“2) The statistical distribution of PFS in copolymer chains (Figure S2), together with low f_{PFS} , prevented the formation of dimers, trimers, and larger PFS sequences, guaranteeing that the relaxation of PFS was governed by the dynamics of the neighboring MMA.”

was revised as:

The similar reactivity ratios between MMA and PFS ensures the formation of a statistic copolymer with random distribution of PFS in the chains (see details in Section 2.3 in the SM). The random structure prevents the PFS units in the copolymer chains from aggregating, resulting in a uniform distribution of P(MMA-sta-PFS) within the blended films, as confirmed by small angle X-ray scattering (SAXS) (Figure S7).

Comment 4:

4) What fraction of the PFS monomers are at the surface? Where are the rest of them?

Responses: On the basis of the water contact angle, we estimated that the areal fraction PFS units at the surface of the films (Fig. 1C) is 25%. The rest of the PFS monomers were buried beneath the surface, which have been clearly demonstrated by our Monte Carlo

simulation in Fig. 1A.

We made the following revisions to the sentence on page 4 to highlight this point:

“The surface concentration of PFS was kept constant (CPFS = 25%; see Figure 1C, and the rest PFS monomers was buried beneath the surface, see Figure 1A)”

Comment 5:

5) Figure 5c: What is the uncertainty in A?

Responses: The standard error of A was roughly smaller than 15%. The errors are small, and the error bars cannot be visually displayed in the figure because they overlap with the data symbol. We included the errors in Table S4 in the SM.

Reviewers' Comments:

Reviewer #1:

Remarks to the Author:

The manuscript has been appropriately revised, and I believe it is now suitable for publication.

Reviewer #2:

Remarks to the Author:

The authors have modified their text based on my recommendations. I have particularly appreciated the new figure in the Supporting information file with a 3D plot of Δ as a function of temperature and frequency, which clearly shows that the increment in Δ observed at surfaces can be uniquely attributed to faster dynamics. I strongly recommend considering this manuscript for publication in Nature Communications.

A small note to consider upon revision of the proofs or the authors would be asked to submit a revised version: I have spotted a typo in the final paragraph, adsorption should read adsorption.

Reviewer #3:

Remarks to the Author:

The authors made good use of the review comments to construct an even stronger manuscript. This reviewer thinks it is now ready for publication. The consensus among the reviewers is correct: this is a strong contribution that will secure many references.